# Pitted-Ground Volcanoes on Mercury

**Ru Xu [1]**, **Zhiyong Xiao [1,2,\*]**, **Yichen Wang [1]** and **Rui Xu [1]**

1   Planetary Environmental and Astrobiological Research Laboratory, School of Atmospheric Sciences,
    Sun Yat-sen University, Zhuhai 519000, China
2   CAS Center for Excellence in Comparative Planetology, Hefei 230026, China
\*   Correspondence: xiaozhiyong@mail.sysu.edu.cn

**Abstract:** On the planet Mercury, pyroclastic deposits formed by explosive volcanism are developed around rimless volcanic pits that are up to dozens of kilometers in diameters. Some pyroclastic deposits on Mercury, however, host no discernable main eruption centers but feature pitted-ground terrains that each consists of many similar sized and irregularly shaped pits. Individual pits are usually much smaller and shallower than typical volcanoes on Mercury. The origin of these landforms is unknown, but it is indicative of styles of volcanism on Mercury and/or post-volcanic modifications. Here, we investigate the possible origin of these peculiar landforms based on their geological context, morphology, geometry, reflectance spectra, and geophysical background. Reflectance spectra of pyroclastic deposits around such volcanoes are comparable with those erupted from typical volcanic pits on Mercury, suggesting a genetic relation between these pitted-ground terrains with explosive volcanism, and the source magma might have similar compositions. Pitted-ground volcanoes are mainly observed in impact structures, and two cases were formed in high-reflectance smooth plains and channeled lava flows. Most pitted-ground volcanoes are relatively degraded compared with typical volcanoes on Mercury, and some might have been formed in geological recent times judged by both their pristine preservation and crosscutting relationship with impact rays. All pitted-ground volcanoes have unconfined morphology boundaries, and each case is composed by dozens of rimless pits that have similar preservation states and interconnected edges. Such morphological characteristics are unique among volcanic landforms on terrestrial bodies, and they cannot be explained by multiple post-eruption collapses of a main explosive volcano. Pitted-ground volcanoes that are developed in lava flows with the same age have different preservation states, suggesting that the pits were not formed by escape of thermally destabilized volatiles from substrate and subsequent roof collapses. The largest pitted-ground volcano ($\sim$3700 km$^2$) is located on the Borealis Planitia, and Bouguer gravity data reveal no larger mass concentration in the subsurface than surrounding terrains, consistent with a paucity of shallow intrusions in the crust of Mercury. Short-term and spatially-clustered explosive eruptions could explain the peculiar morphology and geometry of the pits, suggesting that pits in a given pitted-ground volcano are akin to swarms of monogenetic volcanoes. However, possible magma dynamics for the formation of pitted-ground volcanoes cannot be confirmed until future high-resolution gravity mapping could reveal detailed interior structures beneath these volcanoes. Based on comparative studies with spatially-clustered and similarly aged volcanoes on Earth, we interpret that a combination of pervasive crustal fractures and regional thermal anomaly in the thin mantle of Mercury might have caused such short-term and spatially-clustered explosive eruptions. If this interpretation was true, the heavy degradation state of most pitted-ground volcanoes and the few well-preserved cases are consistent with an overall cooling trend of the mantle, indicating the existence of longstanding heterogeneous thermal structures in the mantle.

**Keywords:** Mercury; MESSENGER; volcanoes; pyroclastic deposits; thermal anomaly

## 1. Introduction

Pyroclastic deposits formed by explosive volcanism are a key window to the understanding of the interior volatile budget [1–4], composition [5], and geodynamics [6–8] of

the mantle of the planet Mercury. Over 130 pyroclastic deposits have been observed on Mercury [9–11], and they occur as thin and diffuse mantling materials that can extend over 140 km from the hypothesized source vents [4]. Targeted observation for a large patch of pyroclastic deposit revealed a relative depletion of elements C and S compared with the surrounding terrain, indicating that oxides of C and S from the juvenile mantle and/or country rocks of magma conduits might be the propellant for the explosive eruption [12]. Pyroclastic deposits on Mercury are likely dispersed by vulcanian-type explosions [3,11], while evidence for shallow intrusions has been elusive [13,14]. Featuring a positive slope of reflectance spectra at the visible to near-infrared wavelengths, most pyroclastic deposits have similar but slightly higher and steeper reflectance spectra than the youngest volcanic plains on Mercury [15–17]. Magma that feeds explosive volcanism on Mercury was most likely sourced from partial melting of the mantle [1], and recent discovery of potential darkish pyroclastic deposits [18] further indicates heterogeneous mantle compositions [5]. Stratigraphic study revealed that pyroclastic deposits were formed at different geological times, with a few as young as the Kuiperian age [18,19]. Younger pyroclastic deposits are less abundant, and they preferentially occur along crustal weaknesses formed by impact cratering and endogenic tectonism [7,11,18–20]. Therefore, the formation frequency of explosive eruptions on Mercury is generally consistent with its thermal history, as a net compressional stress field caused by interior cooling has dominated mercurian lithosphere since ~3.8 Ga, inhibiting the ascending of magma.

Rimless depressions within pyroclastic deposits are interpreted as volcanic vents [1]. The lower surface gravity and the lack of an atmosphere on Mercury would lead to a much wider dispersion of pyroclasts, forming widely distributed pyroclastic deposits without obvious topographic relief [21]. The median sizes of volcanic vents on Mercury are ~13.5 km in the lateral dimension and ~0.6 km in depth [11], while some volcanoes can be as large as ~45 km long [11] and 4 km deep [10]. Nested smaller depressions are frequently visible in many explosive volcanoes, indicating multiple phases of collapses [22] and/or migrated eruption centers [23]. On the other hand, some pyroclastic deposits on Mercury host inside depressions that have peculiar morphology and geometry. Termed as pitted-ground [6] or irregular pitted terrains [11], each such depression consists of several irregularly shaped and rimless pits. The pit size of pitted-ground terrains measured by Jozwiak et al. (2018) is as small as 1 km. Individual pits of pitted-ground terrain are much smaller than typical pyroclastic vents, but they are larger than hollows with spectral relatively blue surrounding haloes on Mercury [6]. Hollows are typically hundreds of meters in scales and flat-floored, and they are likely formed by sublimation of volatiles [24]. Different from typical explosive volcanoes, hollows indicate an active rejuvenation of volatiles in the shallow crust of Mercury [25]. The origin of the pitted-ground terrains in reddish deposits is unknown, and a possible volcanic origin has not been established [26]. Escape of subsurface volatiles that were released by thermal contact with the overlying lava flows and subsequent collapse of the lava roof was interpreted as a possible formative mechanism [6]. This process is somewhat similar with the formation of terrestrial and Martian rootless cones on lava flows [27] and secondary hydroeruptions in pyroclastic flow deposits on the Earth [28]. Martian rootless cones were emplaced over subsurface water ice [27], although rootless cones usually show raised rims. Secondary hydroeruption occurs when hot pyroclastic flows are deposited on a volatile-rich environment, triggering upward flow of gas through fine-grained pyroclastics. Nevertheless, most pitted-ground terrains on Mercury are not developed in discernable lava flows [6,11]. A genetic origin with explosive volcanism is provocative as evidenced by the surrounding reddish and diffusive materials that are consistent with being pyroclastic deposits, and typical volcanic vents are lacking among the pyroclastic deposits. Considering that over 50 such pitted-ground terrains have been reported [6,10,11], their possible origin(s) may contain important information about the diversity of magmatism and thus unrevealed geodynamics of Mercury [26].

No studies has been carried out that focus on these peculiar depressions. Previous research cataloged these peculiar depressions during the investigations of typical eruption

vents. The two previous catalogues do not coincide exactly because of a small difference in morphology identification. Referring to the selection methods employed in previous works, we focus on patches of reddish and diffuse materials that contained pitted-ground terrains in the interior. Cases with sufficient coverages of high-resolution data to investigate their possible origins are highlighted in this work (Section 2.2). We investigate the possible origin of the pitted-ground terrains on Mercury based on their morphology, geometry, geological context, and geophysical background (Section 3). Implications to the mantle dynamics of Mercury are discussed (Section 4).

## 2. Materials and Methods

### 2.1. Data

Various datasets obtained by the MErcury Surface, Space ENvironment, GEochemistry, and Ranging (MESSENGER) spacecraft [29] were used in this study. Data used in this study are available in the public domain via the Planetary Data System. The data IDs are listed in the Table S1 in Supplementary Materials.

For morphological study, monochrome images (central wavelength of 750 nm) that were obtained at various incidence angles by the Narrow-Angle Camera (NAC) and Wide-Angle Camera (WAC) of the Mercury Dual Imaging System (MDIS) [30] were used because of better resolutions. Both single frames (pixel scales as good as 100 m/pixel for WAC images and 15 m/pixel for NAC images) and global mosaics (pixel scales as good as 166 m/pixel) of MDIS images were considered, and the USGS Integrated Software for Imagers and Spectrometers was used to process the data following a standard pipeline with an updated photometric model [31]. We performed a global evaluation of whether or not inside depressions in pyroclastic deposits can be classified as a pitted-ground terrains based on various MDIS WAC global color mosaics, including the enhanced MDIS color mosaic that was built on principle component analyses [31] and locally constructed color mosaics (Table S1 in Supplementary Materials).

Edge-to-floor depths of pitted-ground terrains were measured based on shadow length using the Sun Shadow Tool of the Integrated Software for Imagers and Spectrometers. We employed this semiquantitative method because the elevation points obtained by the Mercury Laser Altimeter (MLA) have scattered spatial distributions. In addition, MLA surface footprint have diameters of 15–100 m, and along-track sampling intervals of ~400 m [32], hence they are not adequate to systematically investigate depths of pitted-ground terrains. Available stereo-pairs and digital elevation models (DEM), e.g., the 665 m/pixel global DEM and those published by Fassett et al. (2016) [33] and Tenthoff et al. (2020) [34] usually have pixel scales much larger than individual pits in the pitted-ground terrains, and they were not usable for our analysis.

To study the reflectance spectra of reddish and diffusive mantling materials associated with pitted-ground terrains, the new eight-band (central wavelengths of 433.2–996.2 nm) MDIS WAC global color mosaic [31] was used (pixel scale of 665 m/pixel). This dataset allows a quantitative comparison of reflectance spectra for nearly all portions of Mercury [31]. To avoid complicated topography and shadows that may affect reflectance spectra, we selected flat and smooth areas and rule out shaded areas for spectra extraction. Reflectance spectra obtained by the Mercury Atmospheric and Surface Composition Spectrometer (MASCS) instrument have a much higher spectral resolution than those extracted from the eight-band MDIS global mosaic [35], and spectral characteristics of global pyroclastic deposits on Mercury (including those around pitted-ground terrains but they were not emphasized for comparison) have been investigated using MASCS data in earlier studies [17].

We referred to the updated global Bouguer gravity [36] and crustal thickness [37] models of Mercury (up to 100 degree) to study the geophysical background of the largest pitted-ground terrain on Mercury. Compared to earlier gravity models, the updated gravity solution HgM008 has substantial improvements in the short- and long-wavelength gravitational field coefficients [36]. Integrating lateral variations in crustal densities, Beuthe et al. (2020) [37] derived a global crustal thickness model of Mercury based on

HgM008 and the global topography. The resolution of the Bouguer gravity and crustal thickness models is up to ~76 km/pixel, and interpolation permits a reduced data record that has a pixel scale of ~10 km (Table S1 in Supplementary Materials). Therefore, the gravity and crustal thickness models can be applied to the largest pitted-ground terrains to investigate the local crustal thickness and mass concentration.

### 2.2. Selection of Research Objects

Definitions for pitted-ground terrains were different in previous works. When cataloging the global inventory of pyroclastic deposits on Mercury, Thomas et al. (2014) [6,10] reported "pitted ground" in reddish pyroclastic deposits (recognized based on false color and morphology) considering the intermediate geometry of the pits between typical mercurian volcanic pits and hollows (e.g., Figure 1a,b). Jozwiak et al. (2018) [11] updated the global inventory of volcanic pits on Mercury and recognized 54 "irregular pitted terrains" in reddish pyroclastic deposits (also recognized based on false color and morphology), which broadly included typical volcanoes that contain nested and/branched small pits (e.g., Figure 1c,d). The two global databases of pitted-ground terrains contain substantial differences due to the different criteria employed (Figure 2). Some of them may be classified as collapsed volcanoes [22] and/or compound volcanoes [23] that were formed due to the migration of eruption centers beneath a major eruption (e.g., Figure 1c,d). In this work, we tightened the definition criteria for pitted-ground terrains to stress their peculiar morphology and geometry compared to typical volcanic pits on Mercury. Here, we consider each pitted-ground terrain consisting of many irregularly shaped and similarly sized rimless pits among a reddish pyroclastic deposit, where no major eruption vent that is substantially larger than the pits is visible (Figure 1e,f). The peculiar morphological criteria used here distinguish pitted-ground terrains from caldera formed by post-eruption collapses, or by migration of eruption centers that are nested within a major outline. Furthermore, we note that reddish and diffusive mantling materials around pitted-ground terrains were usually interpreted as pyroclastic deposits [10,11], but their identity has not been systematically evaluated based on spectral parameters that are representative of typical pyroclastic deposits on Mercury [38].

In total, 14 confirmed pitted-ground terrains are catalogued in this study (Table 1), whose locations are shown in Figure 2. All but one (Figure 1e) of the 14 cases have been cataloged by Thomas et al. (2014) [6,10] and Jozwiak et al. (2018) [11]. It is notable that our definition criteria of pitted-ground terrains are rather conservative. In many cases, clusters of similarly sized pits with interconnected edges are visible on top of a larger but much older volcanic pit (Figure 3a–d). Such younger clusters of pits are not classified yet as confirmed pitted-ground terrains according to our criteria, considering that such pits might be genetically related with the larger but possibly more ancient volcano, e.g., these cases may alternatively be equivalent to compound volcanoes [23]. Additionally, many reddish pyroclastic deposits lack high-resolution and/or high-incidence angle images. For this reason, it is difficult to discern the detailed morphology of their inside volcanoes, although the inside depressions appear to be composed by many pits that have similar sizes and preservation states (Figure 3e,f). Our conservative definition criteria did not deny the possibility for cases with insufficient coverage of high-resolution images, e.g., reddish deposits without clear morphologic data, or similar morphologic terrains that lack obviously reddish deposits. Following the aforementioned criteria, more cases might be confirmed as pitted-ground terrains by future exploration missions.

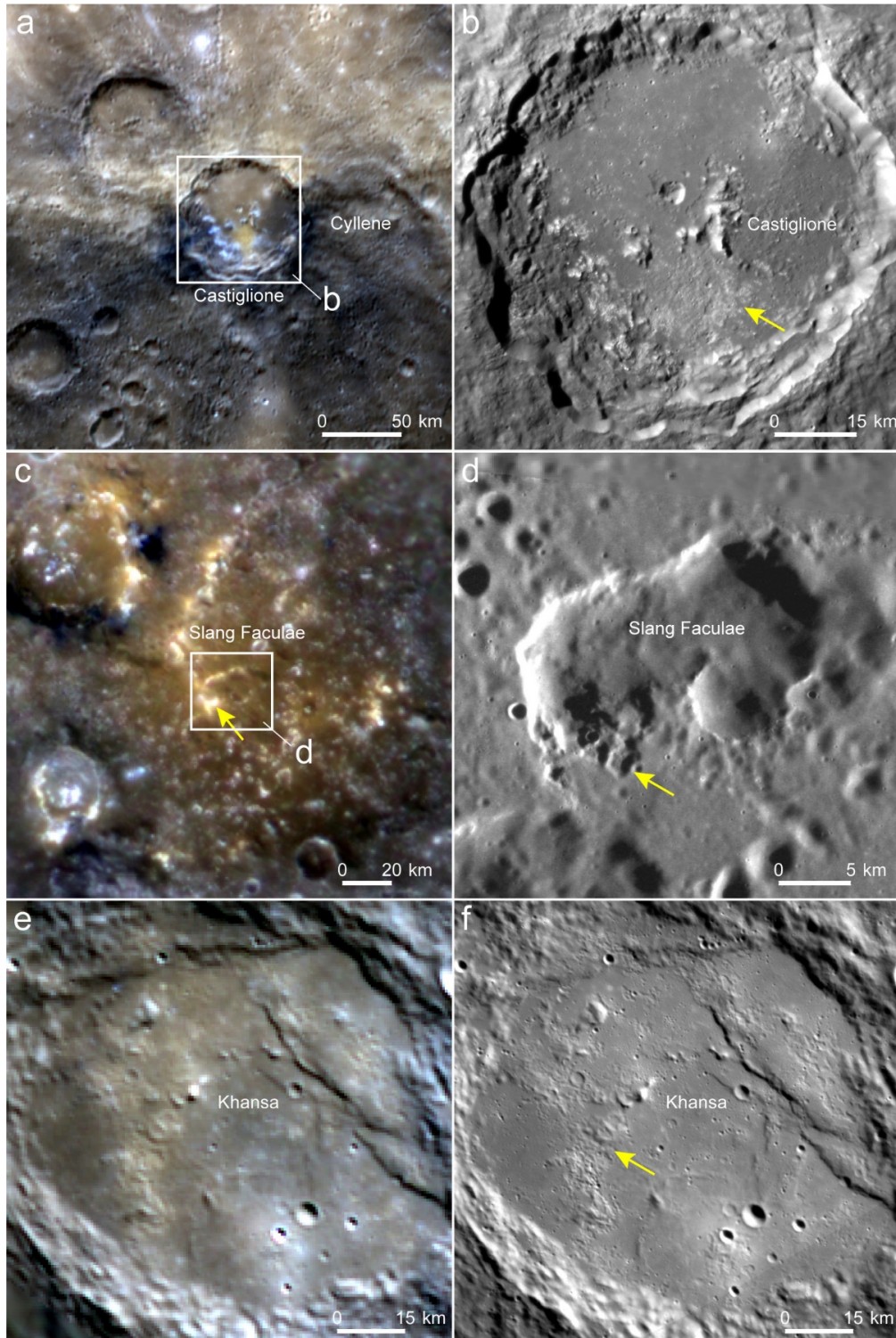

**Figure 1.** Different morphology of volcanic pits on Mercury that are associated with pyroclastic deposits. The first column shows color mosaics, and the second column shows monochrome mosaics for the three examples. (**a**,**b**) A pitted ground defined by Thomas et al. (2014) [6], and the central coordinates of the Castiglione crater are 40.88°S, 87.96°E. (**c**,**d**) An irregular pitted terrain (central coordinates are 24.26°N, 178.97°W) defined by Jozwiak et al. (2018) [11]. (**e**,**f**) A pitted-ground terrain investigated in this study that was not reported before (central coordinates are 59.16°S, 52.56°W). IDs of the base data used are listed in Table S1.

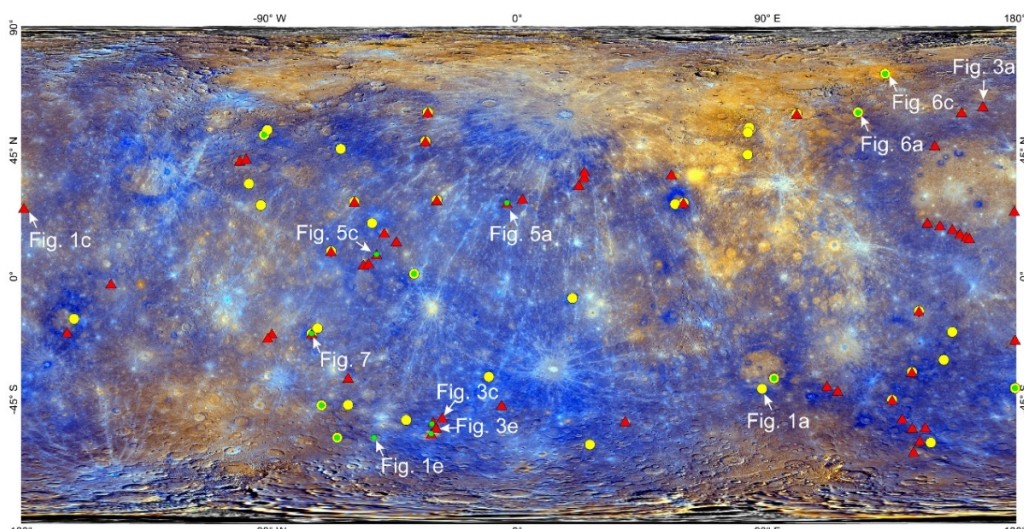

**Figure 2.** Distribution of pitted-ground terrains defined in this study and confirmed as volcanoes, listed by Thomas et al. (2014) [6,10] and Jozwiak et al. (2018) [11]. Yellow spots are those cataloged by Thomas et al. (2014) [6,10]. Red triangles are irregular pitted terrains cataloged by Jozwiak et al. (2018) [11]. Green spots are pitted-ground volcanoes defined in this work. White arrows show locations of cases reported in this study. IDs of the base data used are listed in Table S1.

**Table 1.** Information of pitted-ground terrains that are interpreted as pyroclastic vents in this study.

| Longitude (°) | Latitude (°) | Location | Association with Impact Crater | Reference |
|---|---|---|---|---|
| −74.40 | −21.00 | East of Raphael crater | yes | a c |
| −51.3 | 7.4 | Chaikovskij crater | yes | c |
| −31.70 | −58.09 | Pampu Facula | yes | c |
| −31.43 | −53.86 | Southwest of Sarpa Facula | yes | c |
| −4.20 | 26.20 | Canova | yes | a c |
| 133.07 | 72.82 | Borealis Planitia | no | a b |
| −52.56 | −59.16 | Khansa crater | yes | |
| −90.10 | 50.57 | West of Sholem Aleichem crater | yes | a |
| −37.8 | 0.40 | Southwest of Dominici crater | yes | b |
| −65.58 | −58.70 | Northwest of Rabelais crater | yes | a |
| −70.17 | −48.25 | Northwest of Smetana crater | yes | a |
| 92.32 | −37.33 | Zmija Facula | yes | a |
| 123.96 | 58.61 | Lava channel | no | a |
| 179.5 | −40.85 | Southeast of Liang K'ai crater | yes | b |

[a] Thomas et al. (2014) [6]; [b] Thomas et al. (2014) [10]; [c] Jozwiak et al. (2018) [11].

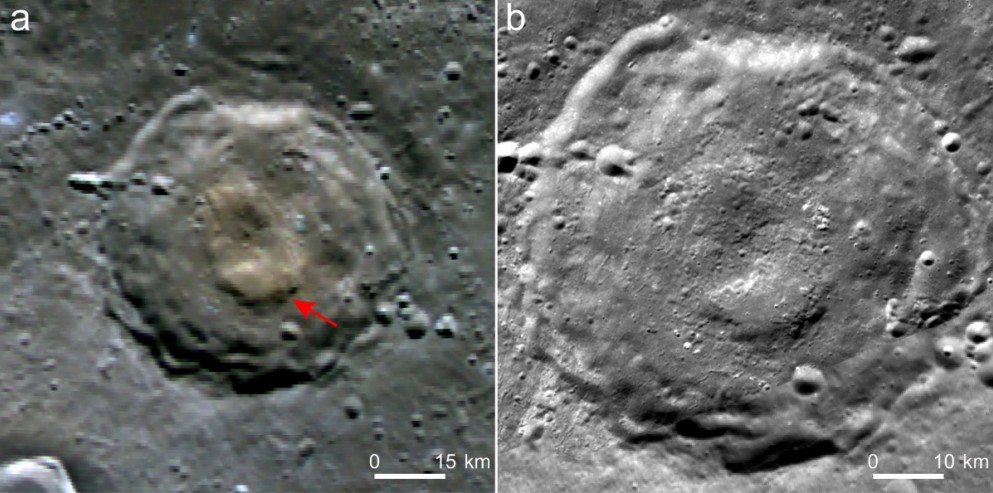

**Figure 3.** *Cont.*

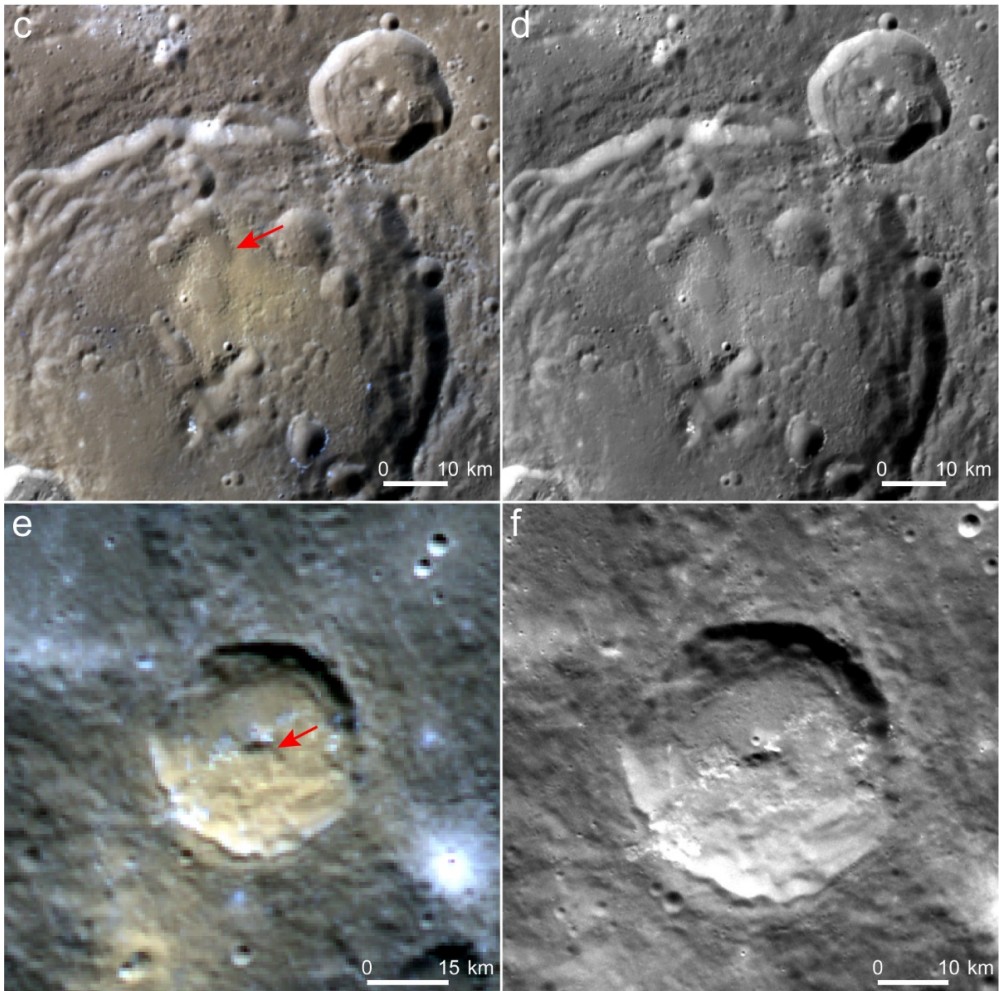

**Figure 3.** Other candidates of pitted-ground terrains that need higher-resolution images to confirm their identity. The first column shows color mosaics, and the second column shows monochrome mosaics for the three examples. (**a**,**b**) A possible pitted-ground terrain that is developed in an older volcanic pit (red arrow) judged by the preservation states (central coordinates are 60.8°S, 168.1°E). (**c**,**d**) A possible pitted-ground terrain that is developed in an older volcanic pit (red arrow) centered at 43.37°N, 82.67°E. (**e**,**f**) A possible pitted-ground terrain among the Bitin Facula (red arrow). IDs of the base data used are listed in Table S1.

## 3. Results

### 3.1. Pyroclastic Deposits around Pitted-Ground Terrains

Characterized by the reflectance at ~750 nm (R750) and spectral slope at visible-to-near-infrared wavelengths (i.e., $\alpha$ = R430/R1000), reflectance spectra of the reddish and diffusive mantling deposits around pitted-ground terrains are comparable with those of typical pyroclastic deposits on Mercury (Figure 4), confirming their identity as pyroclastic deposits [38]. This result of similar reflectance spectra and no obvious absorption feature is visible, is consistent with previous investigations of spectral characteristics of various pyroclastic deposits on Mercury based on MASCS data, which included several cases developed around the pitted-ground terrains cataloged [15,17]. Therefore, we confirm that eruption vents should be within the pyroclastic deposits that are located around the pitted-ground terrains.

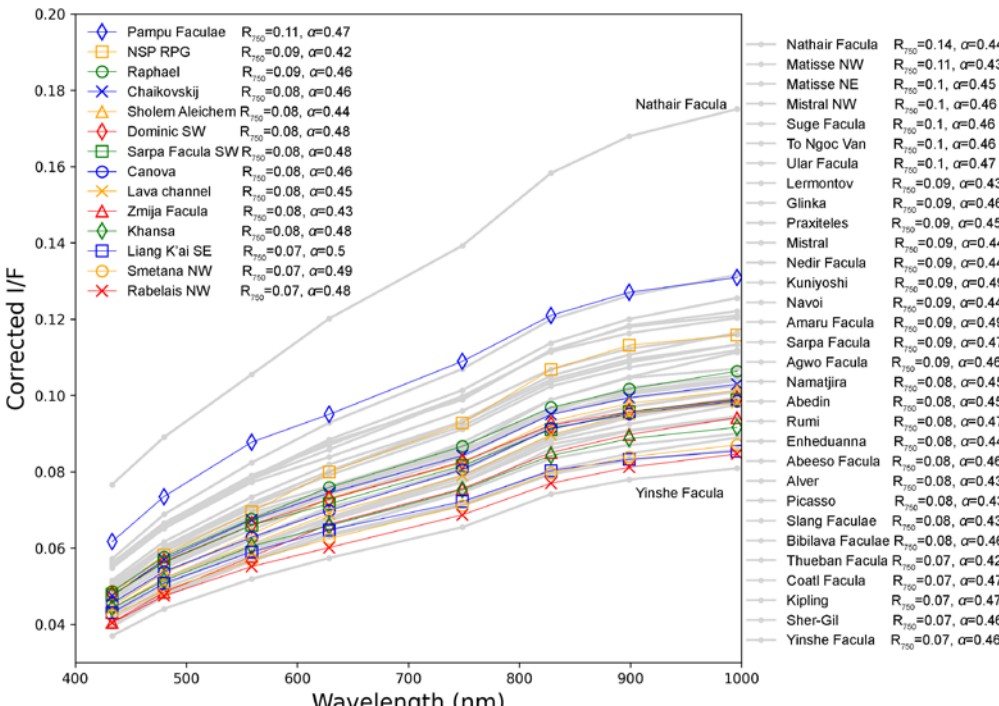

**Figure 4.** Comparison of reflectance spectra of pyroclastic deposits formed by pitted-ground terrains and the other typical explosive volcanoes on Mercury. The spectral data are acquired from the eight-band global color mosaic [31]. Spectral anomalies developed around typical explosive volcanoes marked by grey lines, and those developed around pitted-ground terrains are marked in other colors (see legend). R750 is the reflectance of 750 nm and α is the spectral slope at visible to near-infrared wavelengths as represented by R430/R1000 [38]. The spectra of the other typical pyroclastic deposits on Mercury are sorted up to down by values of R750. The spectral sampling locations are shown in the Figures S1–S7 of Supplementary Materials.

### 3.2. Morphology of Pitted-Ground Terrains

Occurring as shallowly-rugged terrains in reddish pyroclastic deposits, pitted-ground terrains have distinctively different surface textures compared with typical volcanoes and hollows [6,10,11]. Hollows usually have smaller but flatter floors than pitted-ground terrains, and typical explosive volcanoes on Mercury occur as isolated vents that feature major outlines [6,23]. For the pitted-ground terrains, a single patch of reddish pyroclastic deposit can contain several relatively isolated clusters of pits, and pits in a cluster usually have interconnected edges (Figure 5). For individual pits that have discernable edges, their widths are usually measured to be less than ~2 km, and depths of about 0.1 km based on shadow lengths (Table S2 and Figure S8 in Supplementary Materials shows the measurement locations). While individual pits can have comparable sizes with the smallest typical volcanoes [10,11] and large hollows [39] (Table S3), no major volcanic pits in terms of relative sizes are discernable among the pits (e.g., Figure 1e,f). Therefore, pitted-ground terrains are the only discernable candidate pyroclastic vents in centers of the pyroclastic deposits, thus they are interpreted here as pitted-ground volcanoes.

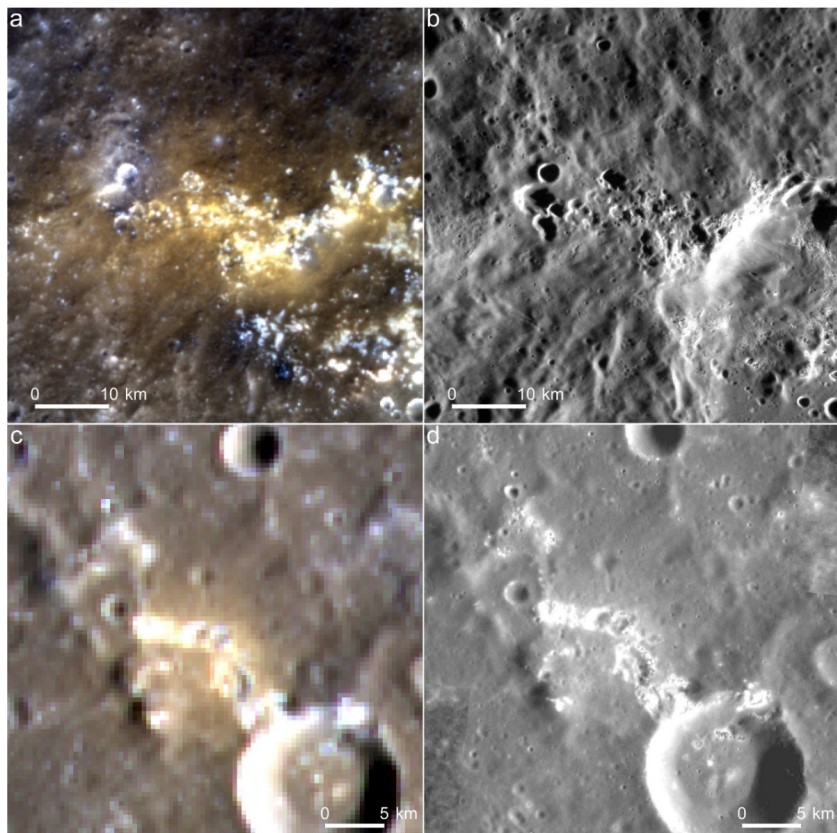

**Figure 5.** Examples of pitted-ground volcanoes on Mercury. The first column shows color mosaics, and the second column shows monochrome mosaics for the two examples. (**a**,**b**) The pitted-ground volcano around the Canova crater. (**c**,**d**) The pitted-ground volcano located in the Chakovskij crater. IDs of the base data used are listed in Table S1.

### *3.3. Pitted-Ground Volcanoes Associated with the Borealis Planitia*

Pitted-ground volcanoes on Mercury do not exhibit spatial clustering at certain latitudes or longitudes, and we found no such cases at the southern high latitudes (i.e., south of 60°S), possibly due to coarser resolution of MDIS images at the southern hemisphere (Figure 2).

While pitted-ground volcanoes are mostly located in floors (Figure 1e) and rims (Figure 5) of impact craters (Table 1; see also Figures S1–S7 in Supplementary Materials), two cases were formed in the northern smooth plains (i.e., Borealis Planitia) and the associated lava channels [40] that were both emplaced by effusive volcanism at ~3.8 Ga [41,42]. Lava channels were located between the Borealis Planitia and Caloris basin, the lava flows buried previous terrains and left residual hills (Figure 6b). The pitted-ground volcanoes were developed in these lava flows with faint bright red haloes (Figure 6a). Compared with the high-reflectance reddish plain materials in the background, the pitted-ground terrain lay on the Borealis Planitia was much rougher in topography (Figure 6d) and faint bright red haloes (Figure 6c) are visible. The surface texture of the two pitted-ground volcanoes is obviously distinct from the background plain materials and lava flows. Pyroclastic deposits associated with the two cases are not obvious as seen in color mosaics due to their similar color with the background high-reflectance red plain materials (Figure 6a–c), but their spectral characteristics as typical pyroclastic deposits have been verified [17]. Pits of the pitted-ground volcano in the Borealis Planitia are much more degraded in topography (Figure 6d) than those of pitted-ground volcano in the lava channels (Figure 6b) as evidenced by the contrasting edge sharpness, although their background terrains were emplaced at the same time [40]. Therefore, the two pitted-ground volcanoes were likely formed at different geological times. This observation suggests that degassing of volatile-

bearing substrate due to contact heating with lava flows and subsequent roof collapse may not be appliable for the origin of the two cases.

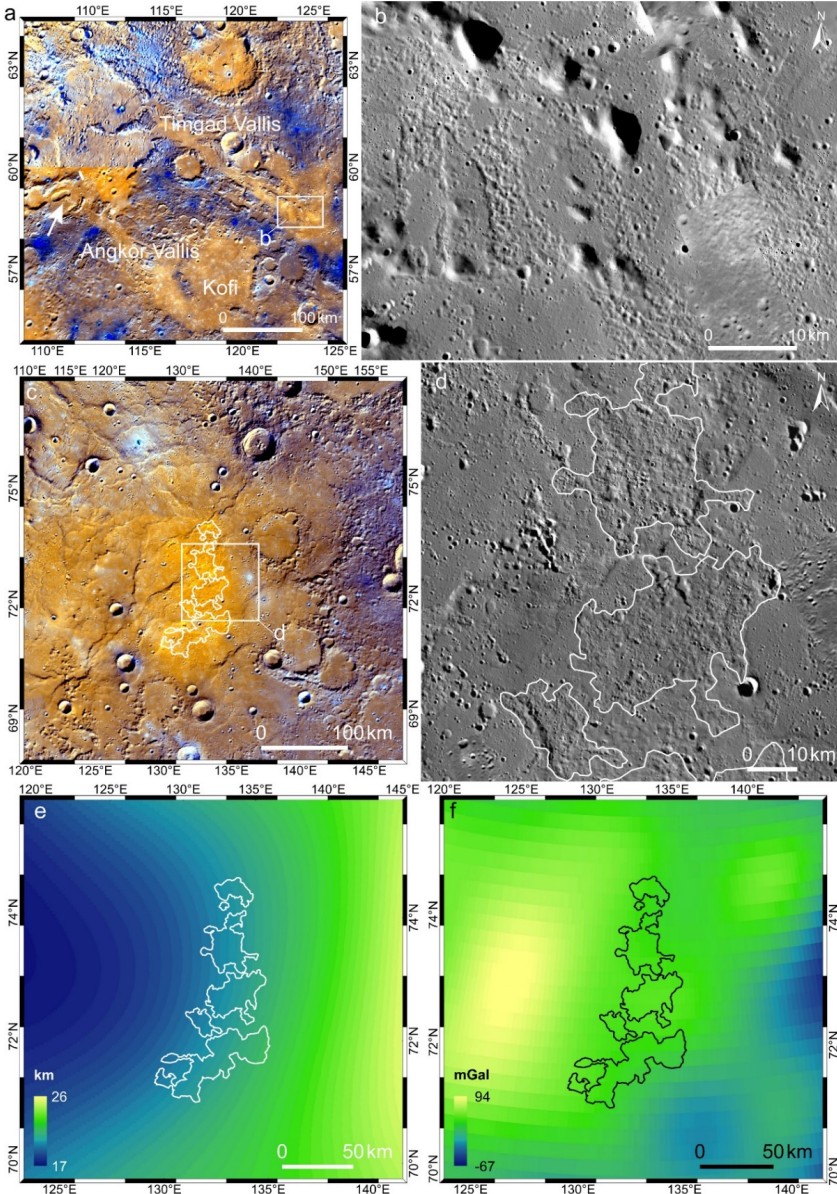

**Figure 6.** Pitted-ground volcanoes developed in high-reflectance plain materials on Mercury. (**a**,**b**) Color and monochrome mosaics for the pitted-ground volcano in the lava channel that is connected with the Borealis Planitia. (**c**–**f**) Color, monochrome mosaics, crustal thickness, and Bouguer gravity anomaly for the only pitted-ground volcano developed in the Borealis Planitia. IDs of the base data used are listed in Table S1.

The pitted-ground volcano in the Borealis Planitia is the largest one on Mercury, covering an area of roughly ~3600 km² (white polygons in Figure 6c). The large size permits a reliable comparison of its crustal thickness (Figure 6e) and Bouguer gravity anomaly (Figure 6f) with the surrounding terrain. Located at the border of the Borealis Planitia and the surrounding highly-cratered terrain, this region, similar to the rest of Borealis Planitia, features a relatively small crustal thickness (~25 km; Figure 6e) compared with the global average (~35 km) [37]. The pitted-ground volcanoes are not associated with distinguishably smaller crustal thickness or larger Bouguer gravity anomaly (Figure 6f) than the background Borealis Planitia. Therefore, no significant mass concentration currently exists beneath this pitted-ground volcano, supporting a mantle derived magma source as

evidenced by the similar spectral characteristics between the pyroclastic deposits and other typical pyroclastic deposits on Mercury [17].

### 3.4. Pitted-Ground Volcanoes in the Raphael Crater

Most pitted-ground volcanoes are heavily degraded as evidenced by their muted morphology (Figures 1f and 6d). Based on crosscutting relationships, we noticed that some pitted-ground volcanoes may be formed in the Kuiperian age, the latest stratigraphic epoch on Mercury [43,44]. Figure 7 shows such a case in a patch of pyroclastic deposit (yellow arrow, Figure 7a) that appears to be superposed on impact rays formed by both the Copley crater to the southwest and an anonymous crater to the northeast (white arrows, Figure 7a). Secondary craters in impact rays are visible outside of but not within the pit floors (white arrows in Figure 7b), suggesting that the pits were likely formed in the Kuiperian age. This interpretation is consistent with crisp morphology of the pit rims, which are in sharp contrast with most similar-sized impact craters in this region (yellow arrows in Figure 7c). Detailed inspection on the morphology further reveals that the pit swarms contain two subgroups that have subtle but recognizable different preservation states, as the more pristine subgroup appears deeper and contain few superposed craters on the pit floors (white arrows; Figure 7d) but the older subgroup appears shallower and contained small craters on the floors (yellow arrows; Figure 7d). The observation here suggests that this pitted-ground volcano may be formed by a prolonged or multiple-phases of eruptions.

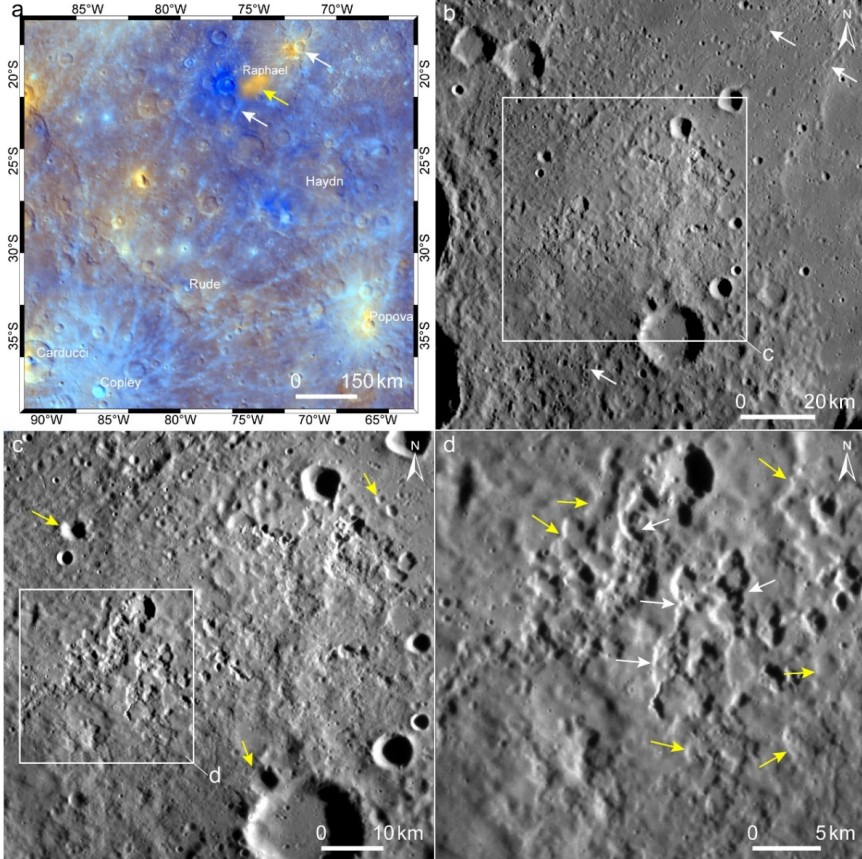

**Figure 7.** A young pitted-ground volcano in the floor of the Raphael crater (D = 342 km; central coordinates are 20.4°S, 283.7°E). (**a**) Color mosaic shows a patch of reddish pyroclastic deposits (yellow arrow) on the southern floor of the Raphael crater. The two white arrows point to impact rays that are formed by the Copley and an unnamed crater. (**b**) Secondaries in impact rays (white arrows) are not visible in the pitted-ground volcano. (**c**) The pits have well-preserved topography, and they are in sharp contrast with the background impact craters (yellow arrows). (**d**) The pits have slightly different preservation states (yellow and white arrows). IDs of the base data used are listed in Table S1.

## 4. Discussion

Pitted-ground volcanoes on Mercury were likely formed by a unique style of explosive volcanism with a cluster of similar-sized irregular rimless pits and thin mantle pyroclastic deposits, considering that such volcanic landforms have no morphological counterparts reported yet on the Moon, Venus, Mars, or Io. Typical pyroclastic deposits on Mercury are dispersed via vulcanian-style volcanism, which were propelled by accumulated volatiles at tips of ascending dikes sourced directly from the mantle [1,11]. If individual pits in a pitted-ground volcano corresponded to spatially-clustered but individually-isolated vulcanian-style eruptions, the similar preservation states of the pits and their frequently-interconnected edges indicate that explosive eruptions occurred at almost the same time. Such an eruption style was not reported before on Mercury. Pits in a given pitted-ground volcano can be classified as a swarm of monogenetic volcanoes, considering that they were formed by qualitatively small-volume eruptions over a brief period of time from a reservoir that has similar compositions [45,46]. Clusters of monogenetic volcanic vents have been noticed on other planetary bodies such as Mars and Earth [46]. However, the geophysical background of monogenetic volcanoes on Mars and Earth is entirely different than that on Mercury, thus each field of monogenetic volcanoes on Mars and Earth was formed with diverse eruption styles [46,47]. Those on Mars are associated with the long-lived Tharsis volcanic province [48] and those on Earth are usually associated with local thermal anomalies in the mantle [49]. If local regions in the mantle of Mercury featured abnormally high thermal activity that has promoted partial melting, e.g., caused by processes such as impact basin formation [50] and heterogeneous distribution of heat production elements in the mantle (analog from the Moon), the local thermal anomaly might be a plausible reason to explain the spatially-restricted small-scale and multiple-center eruptions. This interpretation is generally consistent with the cooling trend of Mercury, since only a few young pitted-ground volcanoes are identified (Figure 7). This interpretation would further indicate that local heterogeneous thermal structures (i.e., warmer regions) might exist in the mantle, and some might have existed in the Kuiperian age.

Unlike vulcanian-style eruptions in lunar floor-fractured craters that are frequently associated with shallow intrusions [51], however, the largest pitted-ground volcano on Mercury is not accompanied by significant igneous intrusions (Figure 6f). Therefore, assuming that the individual pits were also formed by vulcanian-style eruptions, their source magma may also be sourced from the mantle, similar to typical explosive eruptions on Mercury. Following this scenario, the subgroups of pits in a same pitted-ground volcano, which have different preservation states (Figure 7), would indicate that spatially-clustered explosions occurred at different times at the same location. This scenario also indicates that without a shallow magma chamber, multiple dikes that were spaced less than 1 km in distances could simultaneously propagate upward from the partial melting zone in the mantle. We were not aware of the physical plausibility of this scenario, i.e., without a shallow magma chamber, closely spaced swarms of dikes could directly extend from the mantle to the subsurface. We note that the only candidate terrain that may be formed by swarms of dike emplacement on Mercury is the tectonic complex Pantheon Fossae in the center of the Caloris basin [13], which is much larger than the pitted-ground volcanoes. Nevertheless, the formation of pitted-ground volcanoes may be parallel to known styles of volcanism on Mercury. We noticed that several unconfirmed cases of pitted-ground terrains are developed in typical explosive volcanoes on Mercury, suggesting that different styles of volcanic eruptions may occur at a same location and at different times.

While the formation mechanism of the pitted-ground volcanoes is still a mystery in terms of magma dynamics, plausible scenarios need to embrace the current observations, especially about the peculiar morphology and spatial distribution of the pits. The BepiColombo is about to be inserted into the orbit about Mercury in 2025, and the next generation of high-resolution remote sensing data, especially the higher-resolution gravity data to be returned by the Mercury Orbiter Radio science Experiment [52], will improve

observations about these volcanoes (e.g., better geometry control and detailed geophysical background), and the possible origins of these volcanoes should be better resolved.

## 5. Conclusions

Pitted-ground volcanoes on Mercury are composed of many similar-sized and rimless pits that do not contain major volcanic centers. Such volcanoes are unique on both Mercury and the other terrestrial bodies. We applied updated criteria to define pitted-ground volcanoes, noticing that most of them are developed in impact craters. Two pitted-ground volcanoes are discovered in high-reflectance smooth plains and channeled lava flows. Pyroclastic deposits around pitted-ground volcanoes feature similar reflectance spectra with those formed by typical volcanic pits on Mercury, but the peculiar morphology of pitted-ground volcanoes indicates a special formation mechanism. We conclude that pits in these volcanoes are not caused by post-eruption caldera collapses or migration of eruption centers in main volcanoes, and the escape of destabilized volatiles from volatile-rich materials caused by contact-heating of lava flows is not a plausible formation mechanism. Short-term and spatially-clustered explosive eruptions of mantle-derived magma could explain our observations, but the physical plausibility of this scenario warrants further supporting data.

**Supplementary Materials:** The following supporting information can be downloaded at: https://www.mdpi.com/article/10.3390/rs14174164/s1, Table S1: IDs and available address of data used in this study; Table S2: Size and depth of individual pit of pitted-ground volcanoes that are interpreted as pyroclastic vents in this study; Table S3: Differences of hollow, typical eruption vent and pitted-ground volcano; Figures S1–S7: Sampling locations for reflectance spectra of pyroclastic deposits (red polygons) that are developed around pitted-ground volcanoes; Figure S8: Locations of shadow-length measurements (red lines) for the estimation of depths of pitted-ground volcano.

**Author Contributions:** Conceptualization, Z.X.; methodology, Z.X.; software, Z.X. and R.X. (Ru Xu); validation, Z.X., Y.W., R.X. (Ru Xu) and R.X. (Rui Xu); formal analysis, Z.X. and R.X. (Ru Xu); investigation, Z.X. and R.X. (Ru Xu); resources, Z.X. and R.X. (Ru Xu); data curation, Z.X. and R.X. (Ru Xu); writing—original draft preparation, Z.X.; visualization, Z.X. and R.X. (Ru Xu); supervision, Z.X.; project administration, Z.X.; funding acquisition, Z.X. All authors have read and agreed to the published version of the manuscript.

**Funding:** This research was funded by the B-type Strategic Priority Program of the Chinese Academy of Sciences (grant XDB41000000), the National Natural Science Foundation of China (41773063), the Fundamental Research Funds for the Central Universities, and the pre-research Project on Civil Aerospace Technologies (D020201 and D020202) funded by Chinese National Space Administration.

**Data Availability Statement:** Bouguer Gravity Map [36] can be found at https://doi.org/10.1029/2020GL087261; Crustal Thickness Map [37] can be found at https://doi.org/10.1029/2018GL081135; WAC and NAC images of MDIS can be found at https://pdsimage2.wr.usgs.gov/archive/mess-e_v_h-mdis-2-edr-rawdata-v1.0/MSGRMDS_1001/DATA/ (accessed on 21 August 2022); Enhanced Color Global Mosaic can be found at https://messenger.jhuapl.edu/Explore/Images.html#global-mosaics (accessed on 21 August 2022); Eight-band Color Global Mosaic (MDR) can be found at https://messenger.jhuapl.edu/Explore/Images.html#global-mosaics (accessed on 21 August 2022); Monochrome Moderate Solar Global Mosaics (BDR) can be found at https://messenger.jhuapl.edu/Explore/Images.html#global-mosaics (accessed on 21 August 2022).

**Acknowledgments:** The authors are grateful to editor and three reviewers for their helpful reviews.

**Conflicts of Interest:** The authors declare no conflict of interest.

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
