# Peer review of "Pitted-Ground Volcanoes on Mercury"

_remotesensing, doi:10.3390/rs14174164_

Round 1

Reviewer 1 Report

An analysis of pitted ground terrain on Mercury though the lens of volcanic vents.

Although discussion of these deposits in advance of the BepiColombo mission to establish the current state of understanding is valuable, the presentation of this study limits the ability to evaluate the conclusions about the type of volcanic activity on Mercury. The paper needs to be revised to clarify what is new from this study, why the features are selected for study (whether there are others like this out there, or they down selected from available options). I was invited to review based on my experience with volcanism, particularly monogenetic systems, but was unable to evaluate the conclusions based on the lack of detail and clarity provided in text. The major points below highlight the areas that caused this challenge. Additionally line by line notes highlight places in that text that need more detail or better represent the detail needed plus minor typographical errors.

Major items:

1)      There is a challenge in using the term pitted terrain as there have been at least two previous studies using the term using different inclusions criteria. It is therefore very important that the authors be explicit early (and consistently) about what features have been selected for this study and what they hope to add to the knowledge of these features through this study. Currently, the selection of targets is vague up until the method section, and then only further quantified (what red characteristics are used) in the results section. What is needed is clarity on what motivated the study of just these features (and one additional new feature) early and consistently in the manuscript. Providing size ranges of studied features, inclusion criteria, and quantitative bounds on spectral characteristics of the related pyroclastic deposits is needed early on (abstract and introduction). Clear reason for not studying all of the previously recognized features would similarly be helpful.

a.       Related the introduction mentions ‘typical explosive volcanoes on Mercury’ and “hollows” but the distinctive characteristics of these groups relative to pitted ground is not properly mentioned until Line 241, and even then quantitative data (relative sizes that had been mentioned as relevant) are not listed.

b.     The abstract indicates that studied features are pyroclastic deposits that lack the classic vent structure observed elsewhere on Mercury, but do have pitted terrain. This could be reframed as a search for vents considering pitted terrains, rather than a study of pitted terrains in general? If this doesn’t work, the authors must make their pitted terrains unique (via exclusion and inclusion criteria). The abstract is more clear than most of the text, but also would benefit from specific details to demonstrate what supports the conclusions. 

c.       The solution to the previous point and small comments throughout text is a more systematic approach to the layout of the paper. No terms should be used without providing a clear and structured definition (rather than using the term and defining 100 lines later).

2)      Despite being a morphology study there is never a table (in main text or supplement) that covers pit sizes or other dimensions. The distance and spacing of features is mentioned but no data is provided besides images. No shape analysis included. The conclusions based on similarity of sizes and shapes does not stand without data behind it.

3)      The authors do not address other potential formation mechanisms for the pit in any detailed way. Excluding other mechanisms is valuable to supporting the argument in favor of a specific mechanism. Hollows are mentioned, but no discussion of why the features mentioned differ from these structures is presented (besides vague references to size). The abstract says the pits cannot be explained by multiple post-eruption collapses, but this is never discussed in the paper. I also wish to point out relevant other formation mechanisms for the pits that should be considered, and if they wish to argue for a vent interpretation they must exclude, are secondary explosion pits in pyroclastic density current deposits. They briefly mention rootless cones/vents on Mars, but provide no actual discussion of the differences between that mechanism (though lava flows vs. pyroclastic material).

a.       For example, as the authors point out there are other rimless depressions in impact crater that have been interpreted as volatile loss pits (hollows), more information demonstrating that the pitted ground under discussion is distinctive is needed. Just saying they are bigger is insufficient for the argument and should be presented in text, not just as a reference (i.e. line 81)- They call their criteria conservative a comparison with previous criteria (maybe a table) would be really helpful.

b.       References for secondary explosion pits on Earth.

                                                               i.      Gilberston et al. 2020 and references https://www.frontiersin.org/articles/10.3389/feart.2020.00324/full

                                                             ii.      Gabrielli et al. 2020 https://www.researchgate.net/publication/343307869_Geomorphology_and_surface_geology_of_Mount_St_Helens_volcano

Line by line comments:

Line 8 “On the planet Mercury”

Line 176 “an irregular”

Line 195 volcanoes

Method section focuses on selection of targets (which is valuable, though insufficiently clear) and all descriptions of datasets used and spectra collected was in introduction. Please reassign subheadings appropriately. That is put Methods subheading earlier, and the current section title methods should be called selection of targets or similar.

Line 216-227 This section repeats itself. Please rewrite to be more clear.

Line 225: Why should pyroclastic deposits be located in the center of deposits? Demonstrate if this is common on Mercury. Directed jets are not uncommon in pyroclastic eruptions.

Line 241: Not sure the author has properly demonstrated that these features warrant the term volcano for pitted terrain before this point.  (Line 250 attempts this yet appears after first use of phrase).

Line 248: This seems to be an important point of how the studied pits are unique, and yet the only supporting information is visible, not quantitative. This is not very clear.

Line 266: This statement that they are obviously different from background should be state earlier and substantiated with details.

Line 275: than each other? Than the lava?

Line 279: the deposit or the pitted ground morphology is this size?

Line 325: This is an important point that is stated more clearly here than in the upper sections.

Author Response

Thank you for you comments and suggestion. We provide point-by-point responses to the comments as a Word file. Please see the attachment.

Reviewer 2 Report

Strengths:

Conservative definition criteria of pitted-ground terrains.

Weaknesses:

Caption of Figures S1, S2, S3, S4,  S5, S6, S7, refers to Figure 5 instead of Figure 4 (reflectance spectra)

Page 8 line 226: change to “should exist in centers of …”

I liked the authors' presentation style and their unique way of pits' description and clear pointing out of the observed differences.

Author Response

(The authors gave the same response as above.)

Reviewer 3 Report

Dear Editor and Authors,

Thank you for allowing me to review "Pitted-ground volcanoes on Mercury" by Xu and coauthors. The manuscript proposes the investigation of pitted-ground terrains of Mercury of which the origin is still unknown. These terrains share with other volcanic vents the presence of reddish, probably pyroclastic, deposits (also known in nomenclature as faculae). The authors provide an improved description of these features by means of morphological, geometrical, spectral, and geophysical analyses. They select a more conservative set of pitted-ground terrains and conclude that these are of volcanic origin (pitted-ground volcanoes) with magma sources rooted in the mantle and connected to the surface by narrow-spaced dikes. I think this topic is worthy of investigation and that the manuscript is suitable for this journal. However, some moderate implementations are needed. In particular, I am puzzled by the conclusions. While most listed pitted-ground volcanoes are associated with craters, the authors base their conclusions on the only pitted-ground volcano not associated with a crater (Borealis Planitia, where geophysical data are available). I think this is a major weakness of the manuscript's conclusions and I don't feel convinced after reading the manuscript. I am listing below the main issues I found and attaching the manuscript with more punctual comments.

Main issues
1) Materials and Methods: The materials used are pretty well described and listed in supplementary materials. However, by the description provided, it is not yet clear why they needed separate mosaics from the MESSENGER ones. More precisely my questions are:

a) Did the images listed in table S1 provide a better resolution than the available regional MESSENGER-color-mosaics (BDR, LOI, HIE, HIW, MDR, MD3, etc. see https://pds-imaging.jpl.nasa.gov/volumes/mess.html for acronym reference)? Why did you need to use specific MDIS frames, were not the available basemaps enough for some reason?

b) If you produced those mosaics from scratch using the images listed in table S1, did you use a specific photo-correction, or none at all?

2) Table S1: I strongly suggest the use of the standard basemap acronyms (BDR, MDR etc) in the text in order to lower the confusion between PDS-ready products and self-produced products. This would also simplify also the descriptions in table S1, where you could just specify e.g., "self-produced NAC mosaic, self-produced color mosaic, MESSENGER MDR basemap", etc.

I think that Figure 5 is wrongly listed as Figure "4".

3) Pitted-ground terrains/volcanoes: Can you discuss a bit more on the usefulness of being so conservative in the selection of pitted-ground terrains (PGT)? From my understanding of your work, the fact that you are being more conservative with the definition of PGT is important not just because your PGT are actual PGT while the others are not, but just because your kind of study is more reliable and consistent when applied to PGT that share the same morphological and spectral characteristics. If my understanding is right, this should be clearer in the text.

Can all the other previously defined PGT (even those associated with a volcanic vent) be considered pitted-ground volcanoes (PGV) at a different stadium? e.g., in the first stadium you could have the formation of the main volcanic vent, and in a second stadium a PGV forms. Can you discuss (in the Discussion paragraph) a little more on the relationship between your PGV and the previously defined PGT?

4) PGV source proposed scenario: By analysing the geophysical data available for the Borealis Planitia PGV, the authors conclude that "no significant mass concentration currently exists beneath this pitted-ground volcano, supporting a mantle-derived magma source as evidenced by the similar spectral characteristics between the pyroclastic deposits and other typical pyroclastic deposits on Mercury", hence PGV must be connected to the mantle by narrow-spaced dikes. I think that a narrow-spaced set of dikes connected to the mantle could actually provide a significant density contrast at the resolution of the currently available gravity anomaly datasets, especially considering the fact that Borealis Planitia could have crustal-like density given its putative plagioclase-rich composition (e.g., Namur and Charlier, 2017).
Would you consider discussing a second scenario where the source of PGV is shallower, thus of crustal origin? Borealis Planitia is constituted by lava plains of almost the same color of PGT-related reddish terrains, and a major concentration of volatiles could as well trigger multiple explosions.

5) Major weakness: While the majority of PGV are associated with craters, the authors derive their conclusions from the only PGV not associated with a crater (but yet with a large lava infilling):

a) Why are most of these mantle narrow-spaced dikes coincidently associated with impact craters? This is not discussed. These craters are not all large enough to "disturb" the mantle. More likely, it seems that these craters trigger something within the crust, not the mantle. Especially cases like the Canova crater and Chakovskij crater clearly show that PGT are associated with the crater ejecta blanket (i.e., a surficial formation). In Lucchetti et al. (2018) it is observed that also hollows (intertwined with the pyroclastic deposits in the Canova crater case) are triggered by the composition of a more surficial rock formation. If volatiles are responsible for the formation of both features (hollows and PGT), why should one be shallow and the other deep? I think the authors just need to discuss a second scenario of formation for PGV.

b) Why some previously catalogued PGT are on top of larger volcanic vents? What happens to the magma source in this case? Are these cases a totally different thing for the authors? (actually the morphology is very similar). I think this needs to be better discussed.

Author Response

(The authors gave the same response as above.)
